# The effects of idealism and relativism on the moral judgement of social vs. environmental issues, and their relation to self-reported pro-environmental behaviours

Laura Zaikauskaite ®*, Xinyu Chen, Dimitrios Tsivrikos

Department of Clinical, Educational and Health Psychology, Division of Psychology and Language Sciences, University College London, London, United Kingdom

* l.zaikauskaite@ucl.ac.uk

**Data Availability Statement:** All the files are publicly available from the FigShare database, DOI 10.5522/04/12202295.

## Abstract

Many studies have demonstrated that moral philosophies, such as idealism and relativism, could be used as robust predictors of judgements and behaviours related to common moral issues, such as business ethics, unethical beliefs, workplace deviance, marketing practices, gambling, etc. However, little consideration has been given to using moral philosophies to predict environmentally (un)friendly attitudes and behaviours, which could also be classified as moral. In this study, we have assessed the impact of idealism and relativism using the Ethics Position Theory. We have tested its capacity to predict moral identity, moral judgement of social vs. environmental issues, and self-reported pro-environmental behaviours. The results from an online MTurk study of 432 US participants revealed that idealism had a significant impact on all the tested variables, but the case was different with relativism. Consistently with the findings of previous studies, we found relativism to be a strong predictor of moral identity and moral judgement of social issues. In contrast, relativism only weakly interacted with making moral judgements of environmental issues, and had no effects in predicting pro-environmental behaviours. These findings suggest that Ethics Position Theory could have a strong potential for defining moral differences between environmental attitudes and behaviours, capturing the moral drivers of an attitude-behaviour gap, which continuously stands as a barrier in motivating people to become more pro-environmental.

## Introduction

Today's pressing climate change issues call for the search for effective means to inspire people to support environmental movements [1, 2]. So far, academics, including psychologists, sociologists, economists, policy makers and scholars of neighbouring disciplines have employed emotions [3, 4], nudges [5, 6], theories of Planned Behaviour (TPB) [7, 8], Value-Belief-Norm (VBN) [9, 10] and other techniques to understand and foster positive environmental behaviour change. Over recent years, academics have undoubtedly provided a significant amount of insight into barriers to people's adoption of environmentally friendly lifestyles [11–13], yet

**Funding:** This work was funded by UCL Green Impact Team and UCL. The funders had no role in study design, data collection and analysis, decision to publish, or preparation of the manuscript.

**Competing interests:** The authors have declared that no competing interests exist.

many pro-environmental research studies have fallen short of capturing the so-called "attitude-behaviour" gap–the inconsistency between what people say and what they actually do [14, 15].

To address this, scholars have suggested means to target current shortcomings by being cautions on two fronts [14]. First, the methodological front that might encourage rational and socially desirable responses of a typical "good citizen". And second, the theoretical front, which might be overly-reliant on rational decision-making models which underestimate the effects of other determinants, such as social, historical and cultural context, and the complex consumption identities that continue to emerge as a result of people's attempts to meet ever-changing and conflicting social demands [16]. Therefore, newly conceptualised studies aiming to overcome the common drawbacks of pro-environmental research have started to emerge.

For instance, academics have begun linking pro-environmental behaviour to that of moral domain, because moral domain is concerned with what is an acceptable or unacceptable societal norm and serves as an indicator of what is a good or bad, a right or wrong action to take [17–19]. Although it has been argued that pro-environmental behaviours often fail to generate strong moral intuitions because climate change is non-linear (its consequences and their timing are difficult to predict), unintentional (no one planned or wanted it to happen), and is not caused by some clearly identifiable individuals (it's impossible to target the guilty ones who cause the most harm [20]), some studies have found strong support for the influence of a moral dimension [21–23].

For example, research by Dunlap et al. [24], Feygina et al. [25], McCright et al. [26] have suggested that varying moral concerns which stem from the distinct fundamental domains of human morality [27] can lead to highly polarised pro-environmental attitudes between liberals and conservatives, because liberals use harm- and care-based terms, whereas conservatives use purity- and sanctity-based terms when discussing the consequences of environmental degradation. Research by Feinberg and Willer [28] and Wosko et al. [29] found that politicians were more susceptible to environmental messages when the information was presented using terms that were in line with the moral rhetoric of that particular political group.

Similarly, the study by Farrell [30] on why some people, but not others, choose to participate in environmental activism demonstrated the connection between morality and one's tendency to protect the environment. Specifically, the effects of the moral dimension were presented using the concept of sacredness, because sacredness has been theorised to be driven by a powerful moral force which defines prohibitions [31]. In this case, Farrell [30] found that individuals who held the beliefs that nature is a sacred resource were more likely to sign an environmental petition, donate money to environmental causes, and participate in environmental groups because of an increased intrinsic motivation to guard these prohibitions [32]. In contrast, the research on pro-environmental behaviour, moral norms and classical rational choice models, such as TPB and VBN [33–35], yielded inconsistent results, leaving open the question of whether or not morality could be used to predict pro-environmental behaviour using rational choice models.

In this study, we aimed to extend the investigation of the morality and 'attitude-behaviour' gap. More specifically, we set out to explore if any quantitative measures, rather than moral norms [33, 35], could be used to predict environmental actions, and, if so, could these measures be used to determine the difference in moral drivers of environmental attitudes vs. behaviours. We have chosen to meet our aims by using a measure based on moral philosophy, known as the Ethics Position Questionnaire (EPQ) [36]. The EPQ has been demonstrated to have robust effects over a number of morality-related social studies [36–40]; however, its capacity to predict environmental behaviour is not clear. Therefore, the goal of this current study is to find out if (i) individual moral philosophy impacts moral identity and moral

judgement of social issues in a similar manner; (ii) individual moral philosophy impacts moral judgement of social vs. environmental issues in a similar manner; (iii) individual moral philosophy impacts environmental judgement and self-reported pro-environmental behaviours in a similar manner.

## Theoretical background

### Morality and classical behaviour theories

The reason why environmentally friendly behaviours do not appeal to everyone is because they often demand more personal sacrifice than other moral behaviours [41]. In most cases, acting in an environmentally friendly manner requires a significant amount of effort to perform the behaviour or a large investment of resources (e.g. recycling or purchasing solar panels). This is why it was thought that people would be reluctant to change their behaviour to be more sustainable if there was no personal benefit to do so [41]. Following this idea, it became a science-led practice to motivate pro-environmental actions by framing them as financially beneficial. For example, advertisers have often used phrases such as "energy efficient light bulbs" or "saving water saves money", believing that advertised economic aspects would trigger self-interest to perform pro-environmental action [42].

However, the current state of the climate suggests that this approach might not always prove to be effective. Indeed, research studies by Evans et al. [43] and Jia et al. [44] demonstrated how moral motives could be used to impact environmental involvement. Basing their hypotheses on Schwartz's model of social values [45], which suggests that self-interested motives (e.g. power, wealth, achievement) tend to conflict with self-transcendence motives (e.g. benevolence, help, care) that go beyond personal interest [46, 47], the researchers have defined situations where highlighting self-interest motives was more likely to inhibit rather than foster pro-environmental behaviour.

For example, research by Evans et al. [43] found that self-transcendent motives were effective in cases where they aligned with self-transcendent actions, such as pro-environmental behaviour, but did not work in cases where self-transcendent motives were accompanied by self-interest motives that aimed to motivate self-transcendent action, such as "save money by buying more energy efficient light bulbs for the greener planet". Similarly, research by Jia et al. [44] has extended the findings by demonstrating that the effects of self-transcendent vs. self-interest motives were dependent on a person's moral identity (defined as moral values and motivation) [48]. That is, participants (e.g. activists) whose moral identity was centred on self-transcendent values were more likely to be influenced by self-transcendent motives than participants (e.g. non-activists) whose moral identity was centred on self-interest motives. In particular, Jia et al. [44] found that non-activists were more inclined to perform pro-environmental behaviours when the behaviours were framed using self-interest motives, suggesting that pro-environmental action is motivated by the extent to which specific motives are integrated into one's moral identity.

### Morality and classical behaviour theories

In contrast, the research attempting to incorporate a moral element into classical behaviour theories has not always provided clear results. For example, this was the case with studies on the TPB. TPB is a well-established rational choice model that was introduced by Ajzen in 1985 [49]. It allows prediction of psycho-social factors that determine individuals' actions [49, 50]. According to TPB, an individual's behaviour stems from his/her intentions to perform that behaviour. The stronger the intention, the more likely it is that intentions will translate into actions. As such, intentions are predicted by three main components: attitudes, subjective

norms and perceived behavioural control. 'Attitudes' refer to one's positive or negative evaluation of performing certain behaviour; 'subjective norm' refers to behaviour-related opinions of other significant people; whereas 'perceived behavioural control' refers to one's volitional control of that specific behaviour. Therefore, it would mean that people who have positive attitudes, support from family, friends or peers, and perceive themselves as capable of engaging with an action, should have strong intentions of performing that action [7, 51–53].

One of the strengths of TPB is that it allows for the inclusion of additional variables relevant to specific behavioural contexts [54]. In this vein, studies by Kaiser and colleagues [34, 35], or Chan and Bishop [33] have attempted to extend TPB by including moral norms as an additional predictor of intention. Interestingly, support for this idea was found in some environmental behaviour domains, but not in others. For example, Bamberg et al. [55] found moral norms to add value to the TPB framework in predicting willingness to use public transport; however, this did not work in predicting recycling intentions [34, 35]. It was concluded that moral norms did not increase predictive power of the model because there was a lack of discriminant validity between moral norms and attitudes. That is, these two constructs were found to be highly correlated, suggesting that moral norms might already be represented in people's attitudes towards some environmental behaviours, such as recycling.

Unlike TPB, which can be used to predict a broad range of social behaviours [56], Stern and colleagues [57, 58] developed the VBN framework to specifically predict nonactivist environmentalism. The VBN is comprised of 3 key sequential concepts; values (biospheric, altruistic, egoistic), beliefs (ecological worldview, awareness of consequences, ascription of responsibility), and personal norms [59]. The VBN theory assumes that pro-environmental intentions and behaviour are activated by personal norms, which have an already integrated element of morality in them, by assessing one's moral obligation to perform environmental actions [60, 61]. According to VBN, personal norms are dependent on one's values (biospheric, altruistic, egoistic) and beliefs (ecological worldview, awareness of consequences and ascription of responsibility). Therefore, one's ecological worldview is enhanced by biospheric (linked to nature and biosphere), altruistic (linked to welfare of others), but not egoistic (linked to maximising individual benefits) values [58]. Ecological worldview is measured using the "New Environmental Paradigm" and is based on the idea that natural resources are limited, so the abuse of nature should not be acceptable [62]. Ecological worldview directly affects awareness of the consequences and the ascription of responsibility, because it defines what is considered to be harmful, and what detrimental effects could be triggered by not behaving in a pro-environmental manner [63, 64].

However, Kaiser and colleagues [34] critiqued VBN for failing to demonstrate model fit, suggesting that some of its dimensions, including personal norms, lack sufficient integration into the model. In fact, the empirical analysis of the model has found that personal norms served as the only significant variable in predicting several types of environmental behaviours, suggesting that a moral element indeed plays an important role in fostering pro-environmental actions. Despite that, it is less clear why moral norms did not always work in predicting pro-environmental behavior, or what measures could be used to assess these links more consistently.

## The philosophical approach

It appears that the philosophical approach to morality could offer an insight into the question of whether morality is reflected in one's attitudes, beliefs, and values, or whether the judgement of what is moral and what is not stems from one's (in)competence [65, 66]. In this vein, philosophers suggested that moral antecedents of attitudes, beliefs, and values could be explored

using two main philosophical theories, namely deontological and teleological philosophy [67]. These two theories incorporate all branches of philosophical theories, therefore are considered to form the basis of moral philosophy. In principle, deontological philosophy focuses on the role of duty and moral obligation whereas teleological philosophy focuses on the analysis of consequences [68]. Therefore, deontology focuses on taking actions that prevent harm to others, whereas teleology focuses on the morality of consequences, therefore allowing the possibility of harm if positive outcomes outweigh the negative ones.

Based on this approach, Hunt and Vitell [69] developed a General Theory of Marketing Ethics (GTME), which uses deontological and teleological philosophy to explain what polarises people's views on certain ethical situations. According to GTME [69], the triggering mechanism of the model is individual's perception that a situation involves an ethical issue and therefore various alternatives need to be considered for an individual to make a decision based on his/her ethical judgement (the choice that an individual considers to be the most ethical). GTME suggests that the alternatives of possible decisions are evaluated using both deontological and teleological perspectives [70, 71]. Deontological choice is based on the rightness vs. wrongness of the action itself, irrespective of the outcomes [72–74], whereas teleological choice is based on the characteristic that the act is right only if it produces the greater balance of good vs. bad consequences than other available alternatives [75, 76]. Deontologists determine the ethicality of the action based on its consistency or inconsistency with norms proscribing cheating, stealing, deceiving, etc., and prescribing honesty, fairness, justice, etc. [77, 78]. In contrast, teleologists evaluate ethicality based on desirability or undesirability of action's consequences for a particular group, the probability of these consequences, and the importance of that group [68, 79, 80].

The model proposes that ethical judgements may differ from intentions and behaviour because of the inconsistency between teleological evaluation (the most ethical consequences) and the preferred consequences (positive consequences to oneself rather than others) [68]. Therefore, both ethical judgements and intentions should be better predictors of behaviour in situations where the ethical issues are central, rather than peripheral [69, 81]. Although this model has not yet been applied to assess the links between morality and pro-environmental behaviour, it has been proved to be successful in explaining the variance in the (un)ethical behaviour of consumers [82, 83], therefore providing a promising base to capture the effects of morality in pro-environmental domain.

Few studies have attempted to morally extend TPB by incorporating the General Theory of Marketing Ethics as well as moral self-identity within its framework [84, 85]. The striking findings have revealed ethics dimensions, as well as self-identity, to be more significant predictors of ethical behaviour than the original variables of TPB, such as attitudes and subjective norm. However, the generalisability of the findings has received a considerable critique from fellow scholars [66]. First, it's not clear how the findings would apply to everyday life, because the data were drawn from a sample of consumers who were actively engaged with at least one ethical issue of the study, meaning that the test was run on participants who already identified themselves as ethical and demonstrated adequate behaviour. Second, all 3 models (TPB, VBN, GTME) define behaviour as a consequence of attitudes and intentions. The accuracy of this relationship is, indeed, questionable, given that industry reports denote an 'attitude-behaviour' gap by demonstrating that only 3% out of 30% of consumers who define themselves as ethical shoppers reflect their attitudes by consistently making ethical purchases [86].

In summary, the reviewed literature suggests that morality, indeed, plays a significant role in the landscape of pro-environmental decision making, but its effects are not clear [20]. So far, attempts to understand the impact of moral norms through classical behaviour choice models, such as TPB or VBN, did not provide well-defined characteristics of their role in the

environmental domain [34]. The current evidence from existing research implies the idea that the influence of morality could be more subtle [89], therefore defining its effects might require a different grounding. As such, researchers have attempted to capture the impact of the moral dimension by combining TPB with the elements of moral philosophy-based GTME. Despite the promising results, fellow scholars have questioned the applicability of the findings [66], suggesting that models defining behaviour as a consequence of attitudes and intentions might not be well suited to capture the precise impact of the moral dimension, because of the presence of the 'attitude-behaviour' gap.

## Ethics position theory and moral judgement

Another way to measure the differences in people's moral judgement was proposed by Forsyth [36]. Forsyth [37] too has trusted that ethical philosophy serves as the basis on which attitudes, beliefs, and values are being formed. His instrument, the Ethics Position Questionnaire (EPQ), was specifically designed to capture people's sensitivity to wrongdoing [36, 37]. Indeed, studies advocating this idea have provided evidence that ethical ideology alone could explain a significant amount of variance in (dis)similarities between people's moral judgement and their moral conclusions [38, 39, 40]. More specifically, the EPQ was developed to predict a person's analysis of moral issues based on idealistic and relativistic ideologies, which respectively stem from deontological and teleological schools of thought [38, 87]. For this reason, the EPQ measure serves as a generalised two-factor instrument that is designed to assess a person's position on numerous branches of personal philosophies. It was proven to be successful in predicting personal moral judgement of various issues, including ethically controversial social psychology experiments [40], artificial creation of human life [36], treatment of animals [88], morality of business practices [38], reactions to injustice [89], etc.

In that respect, its first dimension, idealism, represents the degree to which individuals believe that moral behaviour will inevitably produce positive consequences [90]. That is, people with high idealistic orientation believe that their decisions should never cause harm to others; desirable consequences can always be obtained with the right action [39, 56]. People in this position were found to be more accurate in identifying if an issue or situation has a moral component, as well as having stronger perceptions of the moral intensity of the issue or situation [91, 92]. In contrast, people with low idealistic orientation tend to believe that undesirable outcomes are sometimes unavoidable; therefore, it's not always possible to avoid harming others even when a person tries to behave in his/her best moral fashion [90].

The second EPQ dimension, relativism, represents the degree to which one rejects the belief that moral decisions should always conform to universal moral principles [39]. People with high relativistic orientation think that one should not rely on moral absolutes at all times. Rather, contextual factors, complexity of the situation, as well as the people involved with the issue should be taken into account when making the final decision [38]. People with high relativistic orientation were found to be less accurate in identifying moral components of a situation or issue, had lower perceptions of the moral intensity of a situation or issue, and demonstrated less social responsibility [37, 39, 93]. In contrast, people with low relativistic orientation were found to believe that moral principles should be followed at all times and that appropriate moral behaviour should vary depending on the context of the situation [89, 90]. Studies using the EPQ measure have demonstrated its robust effects in regression analyses [94, 95] and also its consistency in classifying people into four ethics positions [96].

## Four ethics positions

Forsyth [36, 37] suggested that the EPQ can be used to classify people into four ethics positions by dichotomising the dimensions of idealism and relativism into high and low values to yield a two by two matrix. He termed people holding certain ethics positions as absolutists (high idealism, low relativism), situationalists (high idealism, high relativism), exceptionists (low idealism, low relativism), and subjectivists (low idealism, high relativism). Research experiments have consistently demonstrated that absolutists are highly sensitive to moral components of an issue or situation, and therefore judge immoral actions most harshly [39]. It is believed that their moral judgement is mostly influenced by the principles of deontological philosophy, which is grounded on the principles of moral rightness or wrongness; therefore selfless decision-making is at heart of absolutists [97, 98]. This is reflected by high levels of idealistic attitudes which drive the concern for the welfare of others, as well as low levels of relativistic principles which serve beliefs that universal moral laws must not be violated [68].

In contrast, another highly idealistic position, situationalism, is believed to be inspired by the principles of ethical scepticism [87]. People of this type are sceptical about the necessity to follow universal moral laws, however, they do maintain the belief that the consequences of their actions must never harm others [39]. It has been argued that situationalists believe that moral action is not necessarily good or bad, but "fitting" and this can be reflected by generating the best consequences for the largest number of people [38, 99]. For this reason, moral value of an action is not determined by universal moral principles, but by its consequences to other people.

The next ethics position, exceptionalism, defines the moral profile of people who score low on both idealism and relativism. Unlike absolutists and situationalists, exceptionists are not driven by an idealistic philosophy [36]. People of this type believe that universal moral principles should be followed, however, exceptions could be made if necessary [100]. The moral judgement of exceptionists reflects a more utilitarian decision-making style where harm to others is an acceptable option as long as the positive consequences outweigh the negative ones [101].

Similarly, subjectivists agree with exceptionists that consequences of an action that might sometimes harm others are not always unacceptable [68]. This type too does not support the idea that universal moral principles should always be followed [87]. The decision-making style of subjectivists reflects the philosophy of ethical egoism because people of this type form their decisions based on personal rather than societal gains and losses [39].

## The current study

In the current study, we examined how well EPQ [36, 37] could be used to predict moral identity, moral judgement of social issues, moral judgement of environmental issues, and self-reported pro-environmental behaviours. Although previous research has indicated that moral identity could be used to predict pro-environmental behaviour [44], no research has directly tested whether EPQ could predict moral identity itself, and whether EPQ demonstrates any differences in predicting moral identity vs. other moral variables. For this reason, we have chosen to assess moral judgement of social vs. environmental issues because judgement reflects the attitudes that people have towards common moral scenes. Therefore, we expected to find out if the judgement of social issues is predicted in a similar manner as the judgement of environmental issues, or whether moral attitudes towards social vs. environmental issues differ. Lastly, our final variable, self-reported pro-environmental behaviours, was used to assess whether EPQ could be used to capture the difference between environmental attitudes and self-reported behaviours, denoting the moral reasons of 'attitude-behaviour' gap.

## Method

### Participants and procedures

The experiment was run using Amazon's Mechanical Turk (MTurk) because MTurk participants tend to cover a wider range of ages and social status than the traditional university participant pools, enabling the study to include the responses from a broader sample of US participants [102, 103]. A US sample was chosen because the US constitutes the largest Western population. Therefore, we anticipate the results to be broadly applicable across US participants with similar demographic characteristics.

With regards to the MTurk participant pool, some studies have documented that the use of MTurk is increasing in social science research [104, 105] However, academics often express concerns about data quality and reliability because MTurk participants are anonymous, unsupervised, complete surveys in unknown locations, and, unlike university participants who participate in the research because of the requirements for passing the course or because of an interest in learning about the psychology, are motivated by financial incentives [106]. As a result, some academic studies have previously reported that MTurk participants were found to have issues with fully reading the instructions [106], engaging in distractions such as cell phones [107], or multitasking [106]. However, other recent studies have suggested that MTurkers have done equally well as or outperformed traditional participant pool respondents in factual attention check tasks [108]. In our case, the data demonstrated good internal consistency of the scales, with all Cronbach's alphas higher than 0.60 and factor loadings higher than 0.50, which is considered to be the limit of acceptability [109]. Therefore, the analysis of acquired data demonstrated acceptable level of data reliability.

The questionnaires were distributed using the Qualtrics survey platform. Participants completed the experiment in a web browser and were paid US\$0.94 for participating. The experiment took around 6 min to complete. The University College London Ethics Committee granted ethics approval for this research, and all participants gave written consent. The results were computed using IBM SPSS (v.24).

**Sample demographics.** The final sample (N = 432) consisted of US participants (56% females and 44% males). Most of the respondents were between the ages of 25 and 34 (35%), and 35 and 49 (33%). 40% of the participants were single (never married) and 29% were married (with children). Lastly, 44% of the respondents had a college graduate degree and 54% were employed full time. Table 1 describes the demographic profile of the sample.

### Measures

**Moral philosophy.** Moral philosophy was measured using the Ethics Position Questionnaire (EPQ) that consists of two 10-item scales measuring idealism and relativism [36, 37]. Participants indicated their level of agreement with given statements on a 7 point Likert scale (strongly disagree (1)–strongly agree (7)) with higher scores indicating greater idealism and relativism. The idealism scale included statements such as "People should make certain that their actions never intentionally harm another even to a small degree" and "Risks to another should never be tolerated, irrespective of how small the risks might be". Relativism included statements such as "There are no principles that are so important that they should be a part of any code of ethics" and "What is ethical varies from one situation and society to another". As in previous research [38–40, 68, 87, 94–96, 98], the EPQ dimensions of idealism and relativism revealed a two-factor solution, and Cronbach's α was 0.93 for idealism and 0.88 for relativism.

**Moral identity.** Moral identity was measured using 5-item scale, created by Zhu [110]. This scale was adapted because it conceptualises moral identity in broad terms and this allowed

Table 1. Demographic profile of the sample (N = 432).

| Demographics | Item | N | % |
| --- | --- | --- | --- |
| **Gender** | Male | 191 | 44 |
| | Female | 241 | 56 |
| **Age** | 18–24 | 68 | 16 |
| | 25–34 | 152 | 35 |
| | 35–49 | 142 | 33 |
| | 50–64 | 50 | 11 |
| | 65 and above | 21 | 5 |
| **Marital Status** | Single (never married) | 171 | 40 |
| | Married (no children) | 63 | 14 |
| | Married (with children) | 123 | 29 |
| | Domestic partnership | 35 | 8 |
| | Divorced | 30 | 7 |
| | Widowed | 4 | 1 |
| | Separated | 6 | 1 |
| **Education** | High school or less | 33 | 8 |
| | Some college | 117 | 27 |
| | College graduate | 191 | 44 |
| | Post collegiate | 90 | 20 |
| | None of the above | 1 | 1 |
| **Employment Status** | Full time | 235 | 54 |
| | Part time | 66 | 15 |
| | Self-employed | 47 | 11 |
| | Unemployed | 31 | 7 |
| | Retired | 17 | 4 |
| | Student | 25 | 6 |
| | Other | 11 | 3 |

us to compare whether EPQ could predict such moral identity in the same manner as it could predict the moral judgement of a wide range of social issues. A 7 point Likert scale was used to assess the agreement with items such as "I view being an ethical person as an important part of who I am" or "I am committed to my moral principles", with lower scores representing disagreement and higher scores representing agreement with the statements (strongly disagree (1)–strongly agree (7)). We have obtained Cronbach's α of 0.87.

**Moral judgement.** For the moral judgement task, participants were asked to rate 5 social and 5 environmental issues on a 7 point semantic differential scale with two anchor points (immoral (1)–moral (7)). The social issues were smoking, recreational drug use, casual sex, alcoholism, and abortion, adapted from the study of Zhong [75]. In this paper, moral judgement of social issues will be termed as social judgement, and moral judgement for environmental issues will be termed as environmental judgement. Our aim was to present popular issues that often spark social debate of "what's good and what's bad"; therefore, the social judgement scale included items related to smoking [111], recreational drug use [112], casual sex [113], alcoholism [114], and abortion [115], adapted from the moral judgement study of Zhong [116].

The environmental issues scale was designed by us for the purposes of this study. The popular issues were pollution [117], single use of plastic [118], global warming [119], emissions

[120], and landfill [121]. The composite score for social and environmental issues was created by summing the 5 respective items. Cronbach's α demonstrated good internal consistency, and was 0.78 for social issues and 0.84 for environmental issues. Items for each moral judgement task were presented in a randomised order.

**Pro-environmental behaviours.** Pro-environmental behaviour could be classified as public-sphere behaviour that includes involvement in environmental groups, policy support, etc., and private-sphere behaviour that includes activities such as purchasing, waste disposal, household maintenance, etc. [57, 122]. The focus of our experiment was to study private-sphere behaviours which could be adapted to most households; therefore we adopted 10 pro-environmental behaviour items from Huang's study [122] (Cronbach's α = 0.81). The frequency of performing popular private-sphere pro-environmental behaviours, such as "reducing air conditioning", "buying energy efficient appliances", "composting kitchen waste", etc. was measured on a 7-point Likert scale (never (1)–every time (7)) and presented in a randomised order.

**Control variables.** Three out of five demographic variables (gender, age, education) served as control measures because of their potential relationship with environmental behaviours and both social and environmental judgements. For example, in the previous studies, gender was found to impact environmental concern and behaviours [123–126]. Age and education were controlled because more senior, as well as more educated people were found to possess higher levels of environmental attitudes and therefore have higher willingness to engage in an environmental movement [127, 128]. With regards to moral philosophies, females were found to be more idealistic but less relativistic than males [129, 130], relativism was found to increase with age [129] and also be higher for females with business ethics education [130]. Controlling for the effects that these variables might have on the final outcome allowed us to ensure that the findings reported in this study were not confounded.

Another potential control variable, income [131, 132], was not taken into account because all the purchase related behaviours, presented in Huang's [122] pro-environmental behaviour scale focus on saving money rather than requiring a more expensive investment for performing pro-environmental actions (e.g. "turning off or unplugging electronic devices while not in use", "buying energy efficient appliances", "reducing driving, and walk, bike or use public transportation"). Therefore, we hypothesised that income level should not impact one's frequency of performing pro-environmental actions.

## Results

### Analysis procedures and rationale

We have adopted similar data analysis procedures to those employed in previous studies [94–96, 100] that had assessed the relationships between morality and social or organisational issues using the EPQ [36, 37] scale. First, we ran a principal component analysis to check whether questionnaire items loaded on correct factors, indicating their suitability for further multivariate analysis [133, 134]. Second, to examine the quality of connections, we ran correlation analysis to assess the strength and directionality of the relationships between questionnaire variables [36, 37, 87]. Third, we ran hierarchical multiple regressions to investigate and define the dynamics of proposed relationships between questionnaire variables [89, 94, 95]. Fourth, we used medians as cut-off points and ran an ANOVA to assess the differences between individual ethics positions [39, 96].

### Data analysis

**Normality tests.** Shapiro-Wilk tests were run to test the assumptions of normality. As expected, the results have revealed that the data was normally distributed (see Table 2).

**Table 2. Shapiro-Wilk test results for normality assumptions.**

| Variable | Statistic | df | P |
|---|---|---|---|
| Idealism | 0.971 | 432 | 0.000 |
| Relativism | 0.993 | 432 | 0.043 |
| Moral identity | 0.963 | 432 | 0.000 |
| Moral judgement | 0.985 | 432 | 0.000 |
| Environmental judgement | 0.997 | 432 | 0.000 |
| Environmental behaviours I | 0.987 | 432 | 0.001 |
| Environmental behaviours II | 0.991 | 432 | 0.011 |

*Statistical significance*: P < 0.05

**Principal component analysis.** Principal component factor analysis with a varimax rotation was conducted to validate the measures [109]. Forty-one items out of the original forty-five loaded well on seven factors with a total variance of 61%. As in the study of Davis and colleagues [100], the two last items of relativism loaded on a different factor, and therefore were dropped. Similarly, two items of environmental behaviours were dropped because they did not load on any of the seven factors. Examining the factor loadings for the rest of the environmental behaviour items revealed a natural two-factor solution. Each factor included 4 items with internal consistency estimates of 0.74 and 0.68. We labelled these factors as "environmental behaviours I" and "environmental behaviours II". All the seven factors in the final solution yielded eigenvalues greater than 1.0, factor loadings were greater than 0.50, and consistency estimates exceeded 0.60, which is the limit of acceptability [134]. The results are presented in Table 3 and show key dimensions, items, consistency estimates, factor loadings and communality statistics.

**Pearson correlations.** Tables 4 and 5 present descriptive statistics and a correlation matrix for the variables retained for the further analysis. EPQ results indicate that its two dimensions, idealism and relativism, did not significantly correlate with each other (r = 0.03; P = 0.60) and are orthogonal. This is consistent with previous studies [36, 87]. Idealism significantly correlated with moral identity (r = 0.35; P = 0.000), environmental behaviours I (r = 0.20; P = 0.000), environmental behaviours II (r = 0.20; P = 0.000), and significantly negatively correlated with social judgement (r = -0.26; P < 0.000) and environmental judgement (r = -0.35; P = 0.000). The negative correlations are in line with our expectations because participants were presented with immoral rather than moral issues. In contrast, relativism significantly negatively correlated with moral identity (r = -0.17; P = 0.001) and significantly positively correlated with social judgement (r = 0.21; P = 0.000). No significant correlations with environmental judgement (r = -0.01; P = 0.87) and environmental behaviours I (r = 0.00; P = 0.93) and II (r = 0.01; P = 0.88) were observed. This suggests that moral identity and social judgement were positively related to idealism and negatively related to relativism, while environmental judgement and environmental behaviours I and II were related to idealism but not relativism. As expected, positive correlations were observed between EPQ dimensions and control variables. Specifically, gender (r = 0.25; P = 0.000) and age (r = 0.14; P = 0.004) significantly correlated with idealism, while age (r = -0.26; P = 0.000) and education (r = -0.16; P = 0.001) significantly negatively correlated with relativism.

**Hierarchical multiple regression.** Following the procedures in similar studies [89, 94, 95], hierarchical regression analyses were conducted to test main effects of idealism and relativism as well as their interaction. Separate regression equations were run for moral identity, social judgement, environmental judgement, and environmental behaviours I and II. Control

**Table 3. Factor loadings and communalities.**

| Key dimensions and items | Factor loadings | Communalities |
|---|---|---|
| *Factor 1: Idealism. Cronbach's α = 0.93, Eigenvalue = 6.184, Variance = 15.08%* | | |
| The existence of potential harm to others is always wrong, irrespective of the benefits to be gained | 0.869 | 0.790 |
| One should not perform an action which might in any way threaten the dignity and welfare of another individual | 0.852 | 0.799 |
| Risks to another should never be tolerated, irrespective of how small the risks might be | 0.836 | 0.730 |
| If an action could harm an innocent other, then it should not be done | 0.802 | 0.700 |
| One should never psychologically or physically harm another person | 0.801 | 0.738 |
| People should make certain that their actions never intentionally harm another even to a small degree | 0.772 | 0.685 |
| It is never necessary to sacrifice the welfare of others | 0.737 | 0.615 |
| The dignity and welfare of the people should be the most important concern in any society | 0.683 | 0.597 |
| Deciding whether or not to perform an act by balancing the positive consequences of the act against the negative consequences of the act is immoral | 0.639 | 0.541 |
| Moral behaviours are actions that closely match ideas of the most perfect action | 0.512 | 0.527 |
| *Factor 2: Relativism. Cronbach's α = 0.88, Eigenvalue = 4.473, Variance = 10.91%* | | |
| Questions of what is ethical for everyone can never be resolved since what is moral or immoral is up to the individual | 0.822 | 0.710 |
| Moral standards are simply personal rules that indicate how a person should behave, which should not be applied in making judgments of others | 0.821 | 0.719 |
| Moral standards should be seen as being individualistic; what one person considers to be moral may be judged to be immoral by another person | 0.796 | 0.663 |
| Ethical considerations in interpersonal relations are so complex that individuals should be allowed to formulate their own individual codes | 0.771 | 0.624 |
| What is ethical varies from one situation and society to another | 0.701 | 0.549 |
| Different types of morality cannot be compared as to "rightness" | 0.689 | 0.578 |
| Rigidity codifying an ethical position that prevents certain types of actions could stand in the way of better human relations and adjustment | 0.684 | 0.546 |
| There are no principles that are so important that they should be a part of any code of ethics | 0.519 | 0.456 |
| *Factor 3: Moral identity. Cronbach's α = 0.87, Eigenvalue = 3.719, Variance = 9.07%* | | |
| I am committed to my moral principles | 0.874 | 0.822 |
| I am determined to behave consistent with my moral ideals or principles | 0.866 | 0.810 |
| I am willing to make a sacrifice to be loyal to my moral values | 0.796 | 0.681 |
| I view being an ethical person as an important part of who I am | 0.773 | 0.706 |
| I am willing to place the collective interest over my own personal ego and interest | 0.562 | 0.533 |
| *Factor 4: Environmental judgment. Cronbach's α = 0.84, Eigenvalue = 3.333, Variance = 8.13%* | | |
| Emissions | 0.812 | 0.711 |
| Global warming | 0.775 | 0.611 |
| Pollution | 0.757 | 0.669 |
| Single use plastic | 0.733 | 0.591 |
| Landfill | 0.701 | 0.585 |
| *Factor 5: Social judgment. Cronbach's α = 0.78, Eigenvalue = 2.847, Variance = 6.95%* | | |
| Recreational drug use | 0.787 | 0.698 |
| Alcoholism | 0.738 | 0.611 |
| Smoking | 0.706 | 0.565 |
| Casual sex | 0.693 | 0.590 |
| Abortion | 0.621 | 0.528 |
| *Factor 6: Environmental behaviors I. Cronbach's α = 0.74, Eigenvalue = 2.262, Variance = 5.52%* | | |
| Turn off or unplug electronic devises when not needed | 0.771 | 0.668 |
| Reduce air conditioning | 0.735 | 0.660 |
| Reduce driving, and walk, bike or use public transportation | 0.680 | 0.635 |
| Reduce using plastic bags, or use own bag when shopping | 0.586 | 0.611 |

(*Continued*)

**Table 3.** (*Continued*)

| Key dimensions and items | Factor loadings | Communalities |
|---|---|---|
| *Factor 7*: *Environmental behaviors II. Cronbach's α = 0.68, Eigenvalue = 2.205, Variance = 5.38%* | | |
| Buy energy efficient appliances | 0.723 | 0.576 |
| Recycle newspapers, plastics, cans and glass | 0.683 | 0.585 |
| Compost kitchen waste | 0.611 | 0.510 |
| Buy local products or locally produced foods | 0.598 | 0.576 |
| Total variance = 61.03% | | |
| KMO = 0.879 | | |
| Approx. Chi-Square = 9340.237 | | |
| Df = 820 | | |
| Sig. = 0.000 | | |

variables were entered in Step 1, followed by main effects of idealism and relativism in Step 2, and interaction term for idealism and relativism in Step 3. Both idealism and relativism were centred to reduce multicollinearity [134].

Tables 6 and 7 indicate that some of the control variables (gender, age) were significant predictors, and accounted for 7% of variance in moral identity ($\beta$ = 4.41, P = 0.000), 3% of variance in social judgement ($\beta$ = 4.41, P = 0.000), 2% of variance in environmental judgement ($\beta$ = 3.31, P = 0.000), 2% of variance in environmental behaviours I ($\beta$ = 3.15, P = 0.000), and 2% of variance in environmental behaviours II ($\beta$ = 3.28, P = 0.000).

Despite that, Tables 6 and 7 show that moral philosophies significantly predicted dependent variables over and above the effects of the control variables. Specifically, the addition of idealism and relativism increased the variance by 11% in moral identity ($\beta$ = 4.83, P = 0.000), 8% in social judgement ($\beta$ = 3.88, P = 0.000), 9% in environmental judgement ($\beta$ = 3.01, P = 0.000), 4% in environmental behaviours I ($\beta$ = 3.31, P = 0.000) and 4% in environmental behaviours II ($\beta$ = 3.40, P = 0.000). These effects were driven by idealism and relativism for moral identity ($\beta_{idealism}$ = -0.33, $P_{idealism}$ = 0.000; $\beta_{relativism}$ = 0.11, $P_{relativism}$ = 0.023) and social judgement ($\beta_{idealism}$ = 0.23, $P_{idealism}$ = 0.000; $\beta_{relativism}$ = -0.18, $P_{relativism}$ = 0.000). However, the interaction between idealism and relativism was not significant. In contrast, analyses revealed no main

**Table 4. Descriptive statistics and intercorrelations among study variables.**

|  | Variable | Mean | SD | 1 | 2 | 3 | 4 | 5 | 6 | 7 | 8 | 9 |
|---|---|---|---|---|---|---|---|---|---|---|---|---|
| 1 | **Gender** | 1.56 | 0.50 | | | | | | | | | |
| 2 | **Age** | 2.55 | 1.04 | 0.09 | | | | | | | | |
| 3 | **Education** | 2.79 | 0.87 | -0.07 | 0.12* | | | | | | | |
| 4 | **Idealism** | 4.46 | 1.34 | 0.25** | 0.14** | -0.08 | | | | | | |
| 5 | **Relativism** | 4.00 | 1.20 | -0.09 | -0.26** | -0.16** | -0.03 | | | | | |
| 6 | **Moral identity** | 5.36 | 1.06 | 0.08 | 0.26** | 0.05 | 0.35** | -0.17** | | | | |
| 7 | **Social judgmt.** | 3.50 | 1.29 | -0.12* | -0.16** | -0.01 | -0.26** | 0.21** | -0.25** | | | |
| 8 | **Env. judgmt.** | 3.21 | 1.11 | -0.13** | 0.08 | 0.06 | -0.31** | -0.01 | -0.12* | 0.00 | | |
| 9 | **Env. behs. I** | 4.21 | 1.35 | 0.10* | 0.06 | 0.12* | 0.20** | 0.00 | 0.26** | -0.09 | -0.15** | |
| 10 | **Env. behs. II** | 4.07 | 1.26 | 0.02 | 0.14** | 0.10* | 0.20** | 0.01 | 0.36** | -0.03 | -0.06 | 0.54** |

*P < 0.05;

**P < 0.01; two-tailed

**Table 5. P values of intercorrelations among study variables.**

| | Variable | 1 | 2 | 3 | 4 | 5 | 6 | 7 | 8 | 9 |
|---|---|---|---|---|---|---|---|---|---|---|
| 1 | **Gender** | | | | | | | | | |
| 2 | **Age** | 0.087 | | | | | | | | |
| 3 | **Education** | 0.172 | 0.015 | | | | | | | |
| 4 | **Idealism** | 0.000 | 0.004 | 0.086 | | | | | | |
| 5 | **Relativism** | 0.055 | 0.000 | 0.001 | 0.595 | | | | | |
| 6 | **Moral identity** | 0.086 | 0.000 | 0.293 | 0.000 | 0.001 | | | | |
| 7 | **Social judgmt.** | 0.011 | 0.001 | 0.848 | 0.000 | 0.000 | 0.000 | | | |
| 8 | **Env. judgmt.** | 0.009 | 0.108 | 0.246 | 0.000 | 0.865 | 0.011 | 0.000 | | |
| 9 | **Env. behs. I** | 0.047 | 0.233 | 0.014 | 0.000 | 0.934 | 0.000 | 0.068 | 0.002 | |
| 10 | **Env. behs. II** | 0.733 | 0.003 | 0.098 | 0.000 | 0.881 | 0.000 | 0.579 | 0.254 | 0.000 |

**Table 6. Results of hierarchical multiple regression analyses predicting moral identity and social judgement.**

| | | Model 1 | | | | | Model 2 | | | | | Model 3 | | | | |
|---|---|---|---|---|---|---|---|---|---|---|---|---|---|---|---|---|
| | | $\beta_1$ | $t_1$ | $P_1$ | 95%CI Low$_1$ | 95%CI Up$_1$ | $\beta_2$ | $t_2$ | $P_2$ | 95%CI Low$_2$ | 95%CI Up$_2$ | $\beta_3$ | $t_3$ | $P_3$ | 95%CI Low$_3$ | 95%CI Up$_3$ |
| **Moral identity** | | | | | | | | | | | | | | | | |
| 1. | **Gender** | 0.06 | 1.34 | 0.182 | -0.06 | 0.03 | -0.03 | -0.55 | 0.584 | -0.25 | 0.14 | -0.03 | -0.54 | 0.589 | -0.25 | 0.14 |
| 2. | **Age** | 0.25 | 5.35 | **0.000** | 0.16 | 0.36 | 0.19 | 4.04 | **0.000** | 0.10 | 0.28 | 0.19 | 4.09 | **0.000** | 0.10 | 0.29 |
| 3. | **Education** | 0.03 | 0.54 | 0.592 | -0.08 | 0.14 | 0.04 | 0.84 | 0.404 | -0.06 | 0.15 | 0.04 | 0.84 | 0.401 | -0.06 | 0.16 |
| 4. | **Idealism** | | | | | | -0.38 | -7.13 | **0.000** | -0.33 | -0.19 | -0.33 | -7.12 | **0.000** | -0.33 | -0.19 |
| 5. | **Relativism** | | | | | | 0.11 | 2.28 | **0.023** | 0.01 | 0.17 | 0.11 | 2.40 | **0.017** | 0.02 | 0.18 |
| 6. | **Idealism*** **Relativism** | | | | | | | | | | | 0.04 | 0.89 | **0.374** | -0.03 | 0.07 |
| | **Constant** | 4.41 | 17.56 | **0.000** | 3.92 | 4.90 | 4.83 | 19.47 | **0.000** | 4.34 | 5.32 | 4.82 | 19.43 | **0.000** | 4.33 | 5.31 |
| | $R^2$ | 0.07 | | | | | 0.18 | | | | | 0.18 | | | | |
| | Adj. $R^2$ | 0.07 | | | | | 0.17 | | | | | 0.17 | | | | |
| | $\Delta R^2$ | 0.07 | | | | | 0.11 | | | | | 0.00 | | | | |
| | $\Delta F$ | 11.08 | | **0.000** | | | 27.71 | | | | | 0.79 | | 0.374 | | |
| | ANOVA | $F(3,428) =$ 11.08 | | **0.000** | | | $F(5,426) =$ 18.56 | | **0.000** | | | $F(6,425) =$ 15.59 | | **0.000** | | |
| **Social judgement** | | | | | | | | | | | | | | | | |
| 1. | **Gender** | -0.11 | -2.31 | **0.022** | -0.53 | -0.04 | -0.04 | -0.85 | 0.397 | -0.35 | 0.14 | -0.04 | -0.85 | 0.395 | -0.35 | 0.14 |
| 2. | **Age** | -0.15 | -3.06 | **0.002** | -0.30 | -0.07 | -0.08 | -1.60 | 0.111 | -0.21 | 0.02 | -0.08 | -1.62 | 0.106 | -0.21 | 0.02 |
| 3. | **Education** | 0.00 | 0.02 | 0.988 | -0.14 | 0.14 | 0.01 | 0.14 | 0.892 | -0.13 | 0.15 | 0.01 | 0.13 | 0.895 | -0.13 | 0.15 |
| 4. | **Idealism** | | | | | | 0.23 | 4.78 | **0.000** | 0.13 | 0.31 | 0.23 | 4.79 | **0.000** | 0.13 | 0.31 |
| 5. | **Relativism** | | | | | | -0.18 | -3.70 | **0.000** | -0.03 | -0.09 | -0.18 | -3.71 | **0.000** | -0.30 | -0.09 |
| 6. | **Idealism*** **Relativism** | | | | | | | | | | | -0.02 | -0.47 | 0.639 | -0.08 | 0.05 |
| | **Constant** | 4.41 | 14.21 | **0.000** | 3.80 | 5.02 | 3.88 | 12.41 | **0.000** | 3.27 | 4.50 | 3.89 | 12.41 | **0.000** | 3.27 | 4.50 |
| | $R^2$ | 0.04 | | | | | | | | | | 0.11 | | | | |
| | Adj. $R^2$ | 0.03 | | | | | | | | | | 0.10 | | | | |
| | $\Delta R^2$ | 0.04 | | | | | | | | | | 0.00 | | | | |
| | $\Delta F$ | 5.41 | | **0.001** | | | | | **0.000** | | | 0.22 | | 0.639 | | |
| | ANOVA | $F(3,428) =$ 5.41 | | **0.001** | | | $F(5,426) =$ 10.72 | | **0.000** | | | $F(6,425) =$ 8.95 | | **0.000** | | |

*Statistical significance: P < 0.05 (in **bold**).

**Table 7. Results of hierarchical multiple regression analyses predicting environmental judgement and environmental behaviours I & II.**

| | | Model 1 | | | | | Model 2 | | | | | Model 3 | | | | |
|---|---|---|---|---|---|---|---|---|---|---|---|---|---|---|---|---|
| | | $\beta_1$ | $t_1$ | $P_1$ | 95%CI Low$_1$ | 95%CI Up$_1$ | $\beta_2$ | $t_2$ | $P_2$ | 95%CI Low$_2$ | 95%CI Up$_2$ | $\beta_3$ | $t_3$ | $P_3$ | 95%CI Low$_3$ | 95%CI Up$_3$ |
| **Environmental judgement** | | | | | | | | | | | | | | | | |
| 1. | **Gender** | -0.13 | -2.72 | **0.007** | -0.50 | -0.08 | -0.05 | -1.11 | 0.266 | -0.33 | 0.10 | -0.05 | -1.11 | 0.270 | -0.32 | 0.09 |
| 2. | **Age** | 0.08 | 1.75 | 0.081 | -0.01 | 0.19 | 0.13 | 2.66 | **0.008** | 0.04 | 0.24 | 0.14 | 2.90 | **0.004** | 0.05 | 0.25 |
| 3. | **Education** | 0.04 | 0.78 | 0.437 | -0.07 | 0.17 | 0.01 | 0.29 | 0.769 | -0.10 | 0.14 | 0.01 | 0.32 | 0.747 | -0.10 | 0.14 |
| 4. | **Idealism** | | | | | | 0.32 | 6.63 | **0.000** | 0.18 | 0.34 | 0.32 | 6.75 | **0.000** | 0.19 | 0.34 |
| 5. | **Relativism** | | | | | | -0.01 | -0.29 | 0.772 | -0.10 | 0.07 | 0.02 | 0.38 | 0.702 | -0.07 | 0.10 |
| 6. | **Idealism* Relativism** | | | | | | | | | | | 0.16 | 3.53 | **0.000** | 0.04 | 0.15 |
| | **Constant** | 3.31 | 12.28 | **0.000** | 2.78 | 3.83 | 3.01 | 11.17 | **0.000** | 2.48 | 3.53 | 2.97 | 11.19 | **0.000** | 2.45 | 3.50 |
| | **$R^2$** | 0.03 | | | | | 0.12 | | | | | 0.14 | | | | |
| | **Adj. $R^2$** | 0.02 | | | | | 0.11 | | | | | 0.13 | | | | |
| | **$\Delta R^2$** | 0.03 | | | | | 0.09 | | | | | 0.03 | | | | |
| | **$\Delta F$** | 3.66 | | **0.013** | | | 22.00 | | **0.000** | | | 12.43 | | **0.000** | | |
| | **ANOVA** | $F(3,428) = 3.66$ | | **0.013** | | | $F(5,426) = 11.21$ | | **0.000** | | | $F(6,425) = 11.67$ | | **0.000** | | |
| **Environmental behaviours I** | | | | | | | | | | | | | | | | |
| 1. | **Gender** | 0.10 | 2.09 | **0.037** | 0.02 | 0.53 | 0.06 | 1.12 | 0.264 | -0.11 | 0.41 | 0.06 | 1.12 | 0.262 | -0.11 | 0.41 |
| 2. | **Age** | 0.04 | 0.72 | 0.472 | -0.08 | 0.17 | 0.02 | 0.36 | 0.721 | -0.10 | 0.15 | 0.02 | 0.39 | 0.699 | -0.10 | 0.15 |
| 3. | **Education** | 0.12 | 2.50 | **0.013** | 0.04 | 0.33 | 0.14 | 2.94 | 0.003 | 0.07 | 0.37 | 0.14 | 2.94 | **0.003** | 0.07 | 0.37 |
| 4. | **Idealism** | | | | | | -0.20 | -4.08 | **0.000** | -0.30 | -0.10 | -0.20 | -4.06 | **0.000** | -0.30 | -0.10 |
| 5. | **Relativism** | | | | | | -0.03 | -0.68 | 0.500 | -0.15 | -0.08 | -0.03 | -0.56 | 0.574 | -0.14 | 0.08 |
| 6. | **Idealism* Relativism** | | | | | | | | | | | 0.03 | 0.53 | 0.598 | -0.05 | 0.09 |
| | **Constant** | 3.15 | 9.66 | **0.000** | 2.51 | 3.80 | 3.31 | 9.86 | **0.000** | 2.65 | 3.97 | 3.31 | 9.83 | **0.000** | 2.64 | 3.97 |
| | **$R^2$** | 0.03 | | | | | 0.06 | | | | | 0.06 | | | | |
| | **Adj. $R^2$** | 0.02 | | | | | 0.05 | | | | | 0.05 | | | | |
| | **$\Delta R^2$** | 0.03 | | | | | 0.04 | | | | | 0.00 | | | | |
| | **$\Delta F$** | 3.77 | | **0.011** | | | 8.57 | | **0.000** | | | 0.28 | | 0.598 | | |
| | **ANOVA** | $F(3,428) = 3.66$ | | **0.011** | | | $F(5,426) = 11.21$ | | **0.000** | | | $F(6,425) = 11.67$ | | **0.000** | | |
| **Environmental behaviours II** | | | | | | | | | | | | | | | | |
| 1. | **Gender** | 0.01 | 0.22 | 0.829 | -0.21 | 0.26 | -0.04 | 0.71 | 0.477 | -0.33 | 0.15 | -0.04 | -0.71 | 0.478 | -0.33 | 0.16 |
| 2. | **Age** | 0.13 | 2.77 | **0.006** | 0.05 | 0.28 | 0.12 | 2.47 | **0.014** | 0.03 | 0.26 | 0.12 | 2.47 | **0.014** | 0.03 | 0.26 |
| 3. | **Education** | 0.08 | 1.72 | 0.086 | -0.02 | 0.26 | 0.11 | 2.25 | **0.025** | 0.02 | 0.29 | 0.11 | 2.25 | **0.025** | 0.02 | 0.29 |
| 4. | **Idealism** | | | | | | -0.21 | -4.20 | **0.000** | -0.28 | -0.10 | -0.21 | -4.20 | **0.000** | -0.29 | -0.10 |
| 5. | **Relativism** | | | | | | -0.06 | -1.18 | **0.239** | -0.16 | 0.04 | -0.06 | -1.12 | 0.263 | -0.16 | 0.04 |
| 6. | **Idealism* Relativism** | | | | | | | | | | | 0.01 | 0.18 | 0.855 | -0.06 | 0.07 |
| | **Constant** | 3.28 | 10.82 | **0.000** | 2.69 | 3.88 | 3.40 | 10.90 | **0.000** | 2.79 | 4.01 | 3.40 | 10.88 | **0.000** | 2.78 | 4.01 |
| | **$R^2$** | 0.03 | | | | | 0.07 | | | | | 0.07 | | | | |
| | **Adj. $R^2$** | 0.02 | | | | | 0.06 | | | | | 0.06 | | | | |
| | **$\Delta R^2$** | 0.03 | | | | | 0.04 | | | | | 0.00 | | | | |
| | **$\Delta F$** | 4.03 | | **0.008** | | | 0.04 | | **0.000** | | | 0.03 | | 0.855 | | |
| | **ANOVA** | $F(3,428) = 4.03$ | | **0.008** | | | $F(5,426) = 6.36$ | | **0.000** | | | $F(6,425) = 5.30$ | | **0.000** | | |

*Statistical significance: $P < 0.05$ (in **bold**).

**Table 8. Tests of simple slopes of regression for interaction between idealism and relativism in the prediction of environmental judgement.**

| Relativism: | Low idealism | High Idealism |
|---|---|---|
| **Simple slope** | -0.168 | 0.076 |
| **SE** | 0.062 | 0.050 |
| **t** | -2.700 | 1.523 |
| **P** | 0.007 | 0.129 |
| **95% CI Low** | -0.290 | -0.022 |
| **95% CI High** | -0.046 | 0.175 |

*Statistical significance: P < 0.05

effect for relativism in predicting environmental variables. Rather, idealism alone accounted for variance changes in environmental judgement ($\beta = 0.32$, P = 0.000), environmental behaviours I ($\beta = -0.20$, P = 0.000) and environmental behaviours II ($\beta = 0.21$, P = 0.014). More interestingly, the interaction for idealism and relativism was significant for environmental judgement ($\beta = 3.53$, P < 0.000), but not for environmental behaviours I and II.

To investigate the nature of this interaction, it was probed and plotted using the method recommended by Aiken and West [134], as well as the procedures used in the studies of Henle and colleagues [66] and Winter and colleagues [95]. First, the regression equation was restructured to represent the regression of environmental judgement on centred relativism at low and high levels of centred idealism. Low and high values were computed by adding or subtracting one standard deviation from the centred variable, respectively. Then, simple slopes of the equations were evaluated to determine if they differed from zero (Table 8). The results indicate that there was a significant relationship between relativism and environmental judgement when idealism was lower (b = -0.168, t = -2.700, P = 0.007) but not higher (b = 0.076, t = 1.523, P = 1.029), suggesting that those lower in idealism were associated with higher levels of relativism, and this relation was significant (Fig 1).

**Means differences in ethics positions.** To assess if there were any differences between ethics positions, four groups representing absolutism, situationalism, exceptionalism, subjectivism were formed by using medians of idealism (4.35) and relativism (4.00) as cut-off points [39, 68, 96]. Differences in groups' ethical judgements were assessed for each of the dependent variables using analysis of variance. Table 9 summarises the significant results for moral identity, social judgement, environmental judgement, environmental behaviours I and II.

Table 10 and Fig 2 provide mean ethical judgement scores for all the dependent variables. In line with previous results [68], highly idealistic individuals had the highest level of moral identity ($M_{absolutist} = 5,96$ and $M_{situationalist} = 5.44$ vs. $M_{exceptionist} = 5.13$ and $M_{subjectivist} = 4.93$), and also reported the highest levels of both environmental behaviours I ($M_{absolutist} = 4.42$ and $M_{situationalist} = 4.38$ vs. $M_{exceptionist} = 4.07$ and $M_{subjectivist} = 3.89$) and II ($M_{absolutist} = 4.36$ and $M_{situationalist} = 4.24$ vs. $M_{exceptionist} = 3.84$ and $M_{subjectivist} = 3.76$). In addition, social judgement ($M_{absolutist} = 2.95$ and $M_{situationalist} = 3.48$ vs. $M_{exceptionist} = 3.66$ and $M_{subjectivist} = 3.96$) and environmental judgement ($M_{absolutist} = 2.84$ and $M_{situationalist} = 3.02$ vs. $M_{exceptionist} = 3.70$ and $M_{subjectivist} = 3.35$) were harshest for highly individualistic individuals (Fig 2). Absolutists, followed by situationalists, had the highest level of moral identity, judged social and environmental issues as most immoral, and reported the highest levels of environmental behaviours I & II. Subjectivists reported having the lowest levels of moral identity and were least willing to perform environmental behaviours I and II. In contrast, they judged social but not environmental

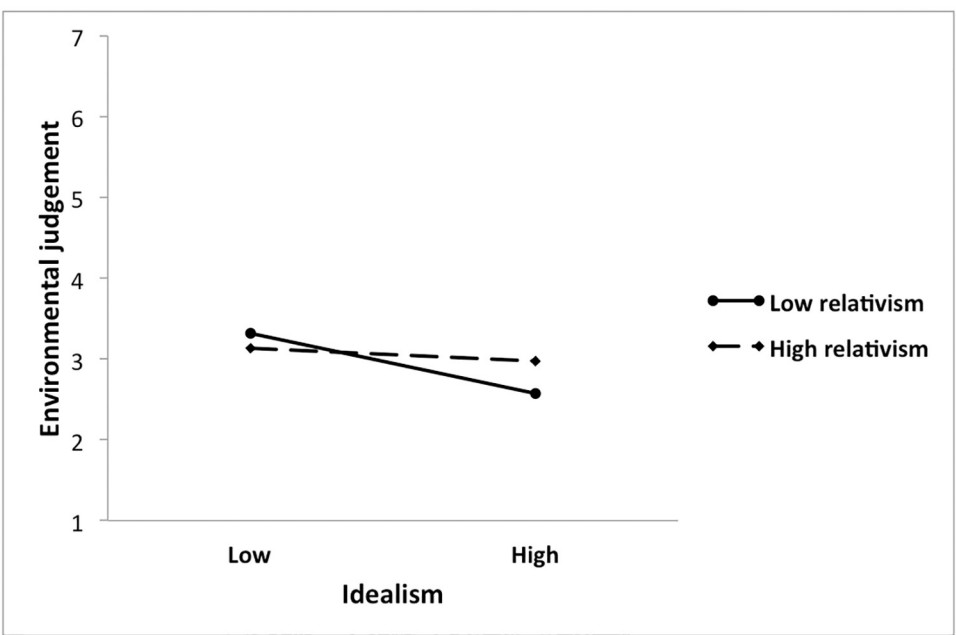

**Fig 1. Interaction between idealism and relativism in predicting environmental judgement.**

issues least harshly. Environmental judgement scores were highest for exceptionists, suggesting that relativism had a different impact on the moral judgement of environmental issues. Over-all, the results are consistent with previous findings indicating that absolutists are the most moral type of all four ethics positions.

## Discussion

In general, these results are consistent with the findings in the reviewed literature. An individual's moral philosophy depicted significant differences in one's moral identity, both social and environmental judgement, and environmental behaviours I and II. Correlational analyses revealed that idealism was correlated with all the dependent variables, but this was not the case for relativism. Specifically, relativism correlated with moral variables such as identity and social judgement, but did not correlate with environmental variables, such as environmental judgement and behaviours I and II. Regression analyses demonstrated that both idealism and relativism significantly contributed to influencing both moral variables. However, the dimensions of idealism and relativism did not interact. People with higher levels of idealism had higher moral identity and judged social issues as more immoral. The opposite pattern was

**Table 9. ANOVA results.**

| Variable | F(3, 404) | P | $\eta_p^2$ |
|---|---|---|---|
| Moral identity | 20.105 | 0.000 | 0.130 |
| Social judgement | 11.349 | 0.000 | 0.078 |
| Environmental judgement | 11.763 | 0.000 | 0.080 |
| Environmental behaviours I | 3.826 | 0.010 | 0.028 |
| Environmental behaviours II | 5.644 | 0.001 | 0.040 |

**Table 10. Mean scores by ethical ideologies.**

|  | Situationalist | | Subjectivist | | Absolutist | | Exceptionist | |
|---|---|---|---|---|---|---|---|---|
|  | *High Idealism High Relativism* | | *Low Idealism High Relativism* | | *High Idealism Low Relativism* | | *Low Idealism Low Relativism* | |
| **N** | 108 | | 107 | | 99 | | 94 | |
|  | M | SD | M | SD | M | SD | M | SD |
| **Moral identity** | 5.44 | 1.00 | 4.93 | 1.04 | 5.96 | 0.77 | 5.13 | 1.17 |
| **Social judgm.** | 3.48 | 1.36 | 3.96 | 1.04 | 2.95 | 1.30 | 3.66 | 1.32 |
| **Env. judgm.** | 3.02 | 1.14 | 3.35 | 1.05 | 2.84 | 1.01 | 3.70 | 1.13 |
| **Env. behs. I** | 4.38 | 1.43 | 3.89 | 1.34 | 4.42 | 1.21 | 4.07 | 1.29 |
| **Env. behs. II** | 4.24 | 1.28 | 3.76 | 1.25 | 4.36 | 1.28 | 3.84 | 1.09 |

found for relativists. That is, people with low levels of relativism had higher moral identity and judged social issues more harshly.

In contrast, idealism but not relativism was found to significantly contribute to influencing environmental judgement. However, the interaction of idealism and relativism was significant. Analyses depicted that individuals low in both idealism and relativism judged negative environmental issues as most moral, whereas those high in idealism and low in relativism judged negative environmental issues as most immoral. These effects became less evident with an increase in idealism.

With regards to environmental behaviours, idealism was the only significant predictor. Specifically, highly idealistic people reported higher environmental behaviours than those who scored lower on idealism. It was interesting to see how the effects of relativism diminished in environmental judgement and disappeared in reported environmental behaviours. Given that relativists reject universal moral principles and tend to perform decisions based on self-interested motives [43], it is not surprising to observe how the strength of relativism changes for variables associated with personal sacrifice [41].

In addition, analysis of variance revealed that variations in all the dependent variables were associated with significant differences in ethical positions. As expected, absolutists, followed by situationalists, represented the most moral type. They had the highest levels of moral identity, judged negative social and environmental issues most harshly, and reported the highest levels of environmental behaviours. The findings are more interesting with regards to exceptionalists and subjectivists. Subjectivists were found to have the lowest levels of moral identity, judge negative social issues most morally, and report the lowest levels of environmental behaviours.

However, this pattern was different for environmental judgement. In this case, exceptionalists were the ones who judged negative environmental issues to be most moral. The differences in which type is the least moral are not unexpected, given that these inconsistencies have already been reported in previous literature [36, 135]. However, these findings depict potentially important differences in decision-making, not only between moral versus environmental variables, but also actual reports of environmental behaviours. The discrepancies between idealism and relativism in predicting environmental judgement and self-reported environmental behaviours suggest that moral philosophy could potentially influence the formation of the "attitude-behaviour" gap [86] and calls for further research to explore these issues.

## General discussion

The present research was designed to examine the role that morality plays in driving environmental attitudes and behaviour. We used a moral-philosophy-based measure, EPQ [36, 37],

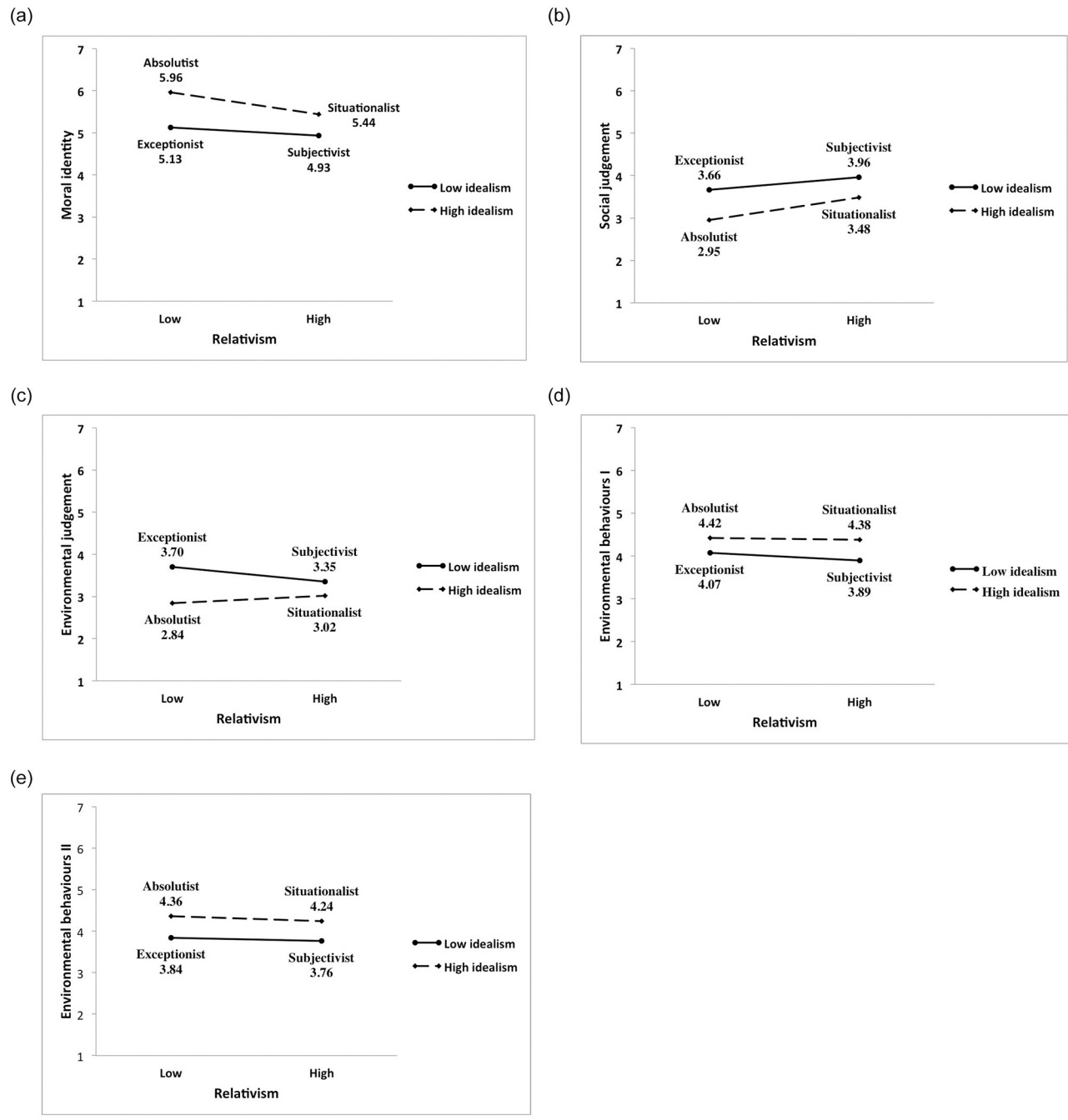

**Fig 2. Mean scores of moral identity (2A), social judgement (2B), environmental judgement (2C), environmental behaviours I (2D) and II (2E) classified per to ethics position type.**

for predicting participants' reactions to moral vs. environmental cases, and compared how its two dimensions, idealism and relativism, are suited to assess the morality of these two distinct types of variables. We found that both idealism and relativism served as strong predictors of moral identity and the moral judgement of social issues. However, this was not the case with environmental variables. Our results revealed that idealism was a strong predictor of both

moral judgement of environmental issues and self-reported pro-environmental behaviours. On the contrary, relativism only interacted with making moral judgements of environmental issues and had no effects on predicting self-reported pro-environmental behaviours. These results suggest that people associate environmental causes with morality; however, the moral drivers of moral vs. environmental variables start to diverge. Moreover, the results portray the reasons behind the 'attitude-behaviour' gap [14, 15, 86], indicating that environmental attitudes could be predicted by idealism and its interaction with relativism, whereas environmental behaviour could only be predicted by idealism.

Therefore, the current research may help explain the reasons behind the 'attitude-behaviour' gap [14, 15, 86]. Previous studies have suggested that morality affects environmental causes [23, 26, 28, 33, 35, 43, 44]; however, the mechanisms through which the moral dimension influences environmental decision-making are not clear [34]. Our results shed light on this debate by indicating that (i) moral and environmental decision-making are not driven by the same moral motives, and that (ii) moral drivers of the moral judgement of environmental issues and self-reported pro-environmental behaviours also differ. Drawing on the findings from this study, it appears that morality is not fully integrated into environmental decision-making, especially when it comes to pro-environmental behaviours. Thus, we believe it is crucial for researchers to distinguish between predictors of attitudes vs. predictors of behaviours, as well as consider whether chosen moral measures are suitable to capture those.

Based on our results, it appears that measures incorporating relativism could better serve in predicting environmental attitudes rather than behaviours. Overall, we argue that moral-philosophy-based measures, such as EPQ [36, 37], might be more suitable for assessing the relationships between morality, attitudes, and behaviours in the environmental domain, and suggest the reasons why studies using other measures, such as moral norms [33, 35], have sometimes failed to find evidence for the hypothesised relationships [34].

For example, studies by Kaiser and colleagues [34, 35] failed to find support for the idea that the moral dimension impacts the formation of environmental attitudes, suggesting that it's likely that morality is already represented in people's attitudes and, therefore, it might be hard to dissociate the two. Given that the study by Chan and Bishop [33] has provided evidence that the moral dimension could be dissociated from attitudes, we suggest that this inconsistency in the results might be because of the measure of morality, which could have fallen short on theoretical grounds and/or because of its susceptibility to social desirability bias [136, 137].

Thus, it might be that measuring morality by means other than moral norms could improve the findings. For example, our results demonstrate the underlying influence of moral philosophies as well as the high reliability of the EPQ as a measure. Combining this instrument with the TPB and VBN could provide an insight into why attempts to add the supplementary variable of moral norms to these models have not always increased their predictive power [34, 35].

Given that our study provides insights into how different ethics positions alter moral and environmental variables, it is possible that splitting participants depending on their ethics position could also influence the predictive capacity of the model and thus allow it to better define what role morality plays in motivating environmental decision-making.

It should be noted that our findings extend a rather limited knowledge base on the impact of the four ethics positions [36, 96, 135], therefore providing additional insight into why certain communication strategies may not appeal to everyone. Within the research on morality and environmental behaviour, for example, Feinberg and Willer [28] and Wosko et al. [29] found that participants were more responsive to environmental appeals when the information

was communicated employing the terms and framing that are often used in their language. Similarly, Jia and colleagues [44] found that aligning internal (e.g., protection) vs. external (e.g., financial) motives to participant's identities can affect pro-environmental choice.

Thus, it seems plausible to expect that communicating moral aspects of environmental behaviours would appeal to highly idealistic ethics positions, whereas communication of the personal benefits could become more effective with an increase in the relativistic dimension.

Future research could seek to explore whether the impact of different communication styles (e.g., highlighting self-transcendent vs. self-interest motives) [43–45] changes across the four ethics positions. In fact, it would be interesting to see if different framing affects both moral and environmental domains in the same way, or whether the effects of moral and environmental domains differ. Additionally, it would be interesting to find out more fundamental reasons why relativism strongly impacted the moral but had a much less effect on environmental domain. Lastly, longitudinal research could explore if observing the environmental behaviour of individuals holding the most moral ethics positions would positively impact the behaviour of those holding less moral ethics positions.

Admittedly, the present study is limited in some key respects. Foremost among these is the notion that we have measured self-reported rather than actual pro-environmental behaviours. We, therefore, cannot make a certain proposition that the findings from our study will correspond exactly to real-life scenarios. An experimental design that includes measuring actual, rather than self-reported behaviours, or at least assessing the experimental variables and self-reported pro-environmental behaviours in two distinct waves (e.g., each wave being a month apart) would help reduce possible response biases (e.g., social desirability) and strengthen the methodological part of the experiment.

Second, we have measured the morality of the negative attitudes towards environmental behaviour but we have not assessed whether participants perceived pro-environmental behaviour as moral. It was previously suggested that moral philosophies have a stronger impact on ethical judgements in cases where the presented issues/behaviours are considered to be highly unethical [68]. It might be that participants did not perceive pro-environmental behaviours as moral, so the effects of relativism were lost. Future studies could attempt to explore if the effects of relativism increase with enhanced moral perception of the issue, or a different group of participants, such as environmental activists.

Third, our study assumes a direct relationship between attitudes and behaviour. The hypothetical nature of the attitude-behaviour relationship has already been critiqued in previous studies [14, 15, 66], suggesting that there might be more factors that impact this relationship. For example, research by Siegel and colleagues [138] demonstrates that immoral attitudes are not as stable as one would expect them to be, whereas research by Crockett and colleagues [139, 140] suggests that the type of deontological vs. teleological judgement could be altered by changing the level of serotonin. Therefore, future research could investigate whether the change in serotonin would lead to a change in participant's ethics position, and would further affect environmental decision-making.

Last, it should be noted that the present findings might not generalise beyond the US population because of the cultural differences between nations across the globe [141, 142]; thus, future research should aim to extend this line of inquiry. Even though additional research must be conducted before we can develop a more specific model of morality and environmental decision-making, we conclude that the present study, indeed, offers new insights into relationships between moral and environmental domains, and suggests new avenues for developing future investigations.

## Conclusion

This study extends previous research by comparing the effects of moral philosophies on moral vs. environmental variables. The above results provide an insight that individual differences in moral philosophies may be an important mechanism to define who will be more inclined to see environmental behaviours as moral and therefore become more engaged in performing environmental actions. Finding communication strategies that appeal to all ethics positions could help to increase environmental behaviours not only for those with high moral standards, but also for individuals who base their decisions on self-interest motives.

## Acknowledgments

We would like to thank Madeleine Stormer for the valuable assistance on this study.

## Author Contributions

**Conceptualization:** Laura Zaikauskaite.

**Data curation:** Laura Zaikauskaite, Xinyu Chen.

**Formal analysis:** Laura Zaikauskaite, Xinyu Chen.

**Investigation:** Laura Zaikauskaite, Xinyu Chen.

**Methodology:** Laura Zaikauskaite, Xinyu Chen.

**Project administration:** Laura Zaikauskaite, Xinyu Chen.

**Supervision:** Dimitrios Tsivrikos.

**Validation:** Laura Zaikauskaite.

**Visualization:** Laura Zaikauskaite.

**Writing – original draft:** Laura Zaikauskaite.

**Writing – review & editing:** Laura Zaikauskaite, Xinyu Chen, Dimitrios Tsivrikos.

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
