## [Decision Letter · Decision Letter 0]

15 Jun 2020

PONE-D-20-12480

The effects of idealism and relativism on the moral judgement of social vs. environmental issues, and their relation to self-reported environmental behaviours

PLOS ONE

Dear Dr. Zaikauskaite,

Thank you for submitting your manuscript to PLOS ONE. After careful consideration, we feel that it has merit but does not fully meet PLOS ONE’s publication criteria as it currently stands. Therefore, we invite you to submit a revised version of the manuscript that addresses the points raised during the review process.

Both reviewers have provided detailed comments and have to be thoroughly accounted for in a revised version.

We look forward to receiving your revised manuscript.

Kind regards,

Nikolaos Georgantzis, Dr.

Academic Editor

PLOS ONE

Journal Requirements:

2. Please ensure that you have included a section outlining your statistical analysis in your Methods section. For details please see here: https://journals.plos.org/plosone/s/submission-guidelines.#loc-statistical-reporting

3. Please include your tables as part of your main manuscript and remove the individual files. Please note that supplementary tables (should remain/ be uploaded) as separate "supporting information" files.

4. Please upload a copy of Figure3, to which you refer in your text on page 22. If the figure is no longer to be included as part of the submission please remove all reference to it within the text.

Reviewers' comments:

Reviewer's Responses to Questions

**Comments to the Author**

1. Is the manuscript technically sound, and do the data support the conclusions?

Reviewer #1: Partly

Reviewer #2: Yes

2. Has the statistical analysis been performed appropriately and rigorously? 

Reviewer #1: Yes

Reviewer #2: Yes

3. Have the authors made all data underlying the findings in their manuscript fully available?

Reviewer #1: Yes

Reviewer #2: Yes

4. Is the manuscript presented in an intelligible fashion and written in standard English?

Reviewer #1: Yes

Reviewer #2: Yes

5. Review Comments to the Author

Reviewer #1: 1. The introduction is a bit out of (academic) focus in certain parts (especially lines 71-90). I believe more needs to be said about moral issues and their importance in order to emphasise the contribution of the paper and the research gap it addresses.

2. In the second section, there are some theories exploring environmental behaviour that have dominated the field of env psychology, env sociology and behavioural sciences. I would encourage the authors to refer to specific authors (or team of authors) in this section.

3. Second section: I believe the authors need to explain why moral value haven't been included in the debates so far.

4. Line 153: Please explain what do you mean by 'positive support'?

5. The authors refer several times to TPB. I think that they need to explain a bit further the specific theory. Same applies also to VBN. The authors assume that the reader is familiar with both frameworks.

6. The authors further down refer to the General Theory of Marketing Ethics. This needs to be explained further. In general I would suggest that the authors re-phrase certain sentences in order to provide more details on the 'content' and key arguments of these theories rather than just referring broadly to them.

7. Section 2.5 is well written and in a way provides what is missing from the literature review (contribution). I would suggest that the authors consider incorporating 2.5 within the literature review.

8. Methods: please explain why you chose participants from US, and why did you choose the specific sampling database. what are the possible negative implications of this choice.

9. Methods: The authors refer to the characteristics of the sample but considering how the sample was selected how can they compare it with a real population? do the authors assume that the results of this study are not possible to be generalised?

10. The selection of the questionnaire variables needs to be better justified. why these questions, why these scales?

11. Line 392. Please clearly state which variables. Is income one of them? If not, why not as I believe it is often connected with env behaviour

12. I might have missed that but what software did you use? A short description of the process of analysing the data would be useful (e.g. why you used Principal Component Analysis)?

13. Line 510-520. I suppose this information could be on a Table

14. the discussion is more like a Summary of the results rather than a discussion. I believe it needs to be rewritten and possibly merged with implications.

15. The implications section lacks references. I believe you need to place your findings in relation to the wider literature. This will help you emphasise the contribution of the paper.

Other comments:

1. Line 66 parenthesis missing at VBN

Reviewer #2: (38-53) Abstract is written in such a way that it is unclear. At first sight it is not straightforward to grasp which of the findings/methods mentioned already existed (and so serve as a motivation) and which ones refer to the present study. A more consistent use of past continuous and present perfect, as well as, the specification of the actor of some sentences would help clarify this.

Introduction could benefit from a final paragraph explaining how the rest of the report is laid out.

(71-72) Comment: This claim seems somewhat stretched. Perhaps rephrase to label it as "one of the most important challenges". Also, the word disbelief seems unsuitable. Perhaps, "difficulty to grasp" or similar would be less of an "accusation".

(83-85) Comment: No-one is specifically to blame but current societies have been created based on consumption patterns and culture which results in unsustainable use of resources and generation of environmental impacts. While "no-one" is to blame, we really are all to blame since we all are part of the responsible system. The argument is still valid, however I suggest the authors re-framed along the lines of: this idea (environmental impacts as a consequence of consumption implied by current consumption practices), although common in science, is not easy to make mainstream - therefore perpetuating the effects of its ignorance in individual consumers. See: Reisch, L.A. and Thogersen, J. eds., 2015. Handbook of research on sustainable consumption. Edward Elgar Publishing.

(96-97) It seems to place all of the weight of consumer-cultural change on the consumer. This seems to disregard the fact that the institutional context also informs and affects people's consumption behaviour. While consumer behaviour is important in understanding the factors behind environmental behaviour, consumers alone are unable to change their consumption culture without the right external (institutional) support. Therefore, the abuse of natural resources cannot be attributed solely to the consumer (or their morality). Although a similar argument can still be made, that the degree of engagement in nature-protection is different and this may partly have to do with issues of moral judgements. I suggest that the authors reword it in order to reflect the issue explained above. See: Reisch, L.A. and Thogersen, J. eds., 2015. Handbook of research on sustainable consumption. Edward Elgar Publishing. See: Reisch, L.A. and Thogersen, J. eds., 2015. Handbook of research on sustainable consumption. Edward Elgar Publishing.

(340-341) The choice of participant recruitment platform is justified by claiming that the distribution better reflects the general population. This seems to be assumed, can this be supported with evidence? Also, do the authors mean "the general population of the US" or the global population? This should be coherent with the fact that participants are from the US.

(355 Measures) This section contains some very superficial explanations of some measures for example:

(389-391) "Environmental behaviours": Some examples of the type of items used could be added here to quickly illustrate their nature. Why was this scale chosen for measuring environmental behaviours? This is not trivial and requires some rationale. Another example is

(395-397) "Control Variables": Do the authors mean that gender, age and education are the 3 variables they controlled for? This seems unclear at first glance.

The "Results" section looks fine. However, the "Discussion" section seems redundant as it basically just summarises the Results section more comprehensively. Perhaps the "Discussion" section could be streamlined and absorbed into the "Results" section.

The "Implications" section is good and, I believe, could just replace the "Discussion" section. However, there is a lack of consideration for limitations of the research. Some examples are: the use of self-reported methods (social desirability bias and potentially hypotheticallity bias), the use of respondents only from the US and how this may affect generalisability of the results and, when measuring the effect of moral philosophies on environmental behaviour, you are effectively measuring an attitude-behaviour relationship which may be subject to similar problems to the TPB that result in the att-bh gap (following the authors' argumentation from lines 320-321).

6. PLOS authors have the option to publish the peer review history of their article (what does this mean?). If published, this will include your full peer review and any attached files.

Reviewer #1: No

Reviewer #2: No

---

## [Author Response · Author response to Decision Letter 0]

29 Jul 2020

Reviewer #1

1.1. The introduction is a bit out of (academic) focus in certain parts (especially lines 71-90). I believe more needs to be said about moral issues and their importance in order to emphasise the contribution of the paper and the research gap it addresses.

1.1. Response: The introduction has now been rewritten and focuses on morality and the ‘attitude-behaviour’ gap, rather than pro-environmental issues in general.

1.2. In the second section, there are some theories exploring environmental behaviour that have dominated the field of env psychology, env sociology and behavioural sciences. I would encourage the authors to refer to specific authors (or team of authors) in this section.

1.2. Response: We have edited all the sections and referred to specific authors where possible.

1.3. Second section: I believe the authors need to explain why moral value haven't been included in the debates so far.

1.3. Response: We have made this section more specific [137-174], and have contextualised the research on self-interest and self-transcendent values.

1.4. Line 153: Please explain what do you mean by 'positive support'?

1.4. Response: This sentence has been deleted, as it no longer fitted within the edited text.

1.5. The authors refer several times to TPB. I think that they need to explain a bit further the specific theory. Same applies also to VBN. The authors assume that the reader is familiar with both frameworks.

1.5. Response: Further explanations have been added [177 – 192; 206 - 225]. 

1.6. The authors further down refer to the General Theory of Marketing Ethics. This needs to be explained further. In general I would suggest that the authors re-phrase certain sentences in order to provide more details on the 'content' and key arguments of these theories rather than just referring broadly to them.

1.6. Response: The GTME has been explained in more detail and we have added more ‘content’ to the theories [251 – 269]. 

1.7. Section 2.5 is well written and in a way provides what is missing from the literature review (contribution). I would suggest that the authors consider incorporating 2.5 within the literature review.

1.7. Response: This section has now been incorporated within the lit. review [297 - 310].

1.8. Methods: please explain why you chose participants from US, and why did you choose the specific sampling database. what are the possible negative implications of this choice.

1.8. Response: Explanation added [416 - 439].

1.9. Methods: The authors refer to the characteristics of the sample but considering how the sample was selected how can they compare it with a real population? do the authors assume that the results of this study are not possible to be generalised?

1.9. Response: We have added this comment to the limitations section [870 – 877].

1.10. The selection of the questionnaire variables needs to be better justified. why these questions, why these scales?

1.10. Response: The explanation for the questionnaire variables added [471 – 530].

1.11. Line 392. Please clearly state which variables. Is income one of them? If not, why not as I believe it is often connected with env behaviour

1.11. Response: We have clarified which variables and have added an explanation regarding income [510 – 511; 523 - 530].

1.12. I might have missed that but what software did you use? A short description of the process of analysing the data would be useful (e.g. why you used Principal Component Analysis)?

1.12. Response: We have added the software [444 – 445], and a short rational of the analysis procedures [534 – 545].

1.13. Line 510-520. I suppose this information could be on a Table

1.13. Response: This information is on Table 9 [689]. However, we have chosen to pick and articulate the key values to emphasise their importance and effects.

1.14. the discussion is more like a Summary of the results rather than a discussion. I believe it needs to be rewritten and possibly merged with implications.

1.14. Response: This section has now been placed after the results section [695], and we have added a general discussion section to present the findings in relation to the wider literature [746].

1.15. The implications section lacks references. I believe you need to place your findings in relation to the wider literature. This will help you emphasise the contribution of the paper.

1.15. Response: See comment 1.14.

Other comments:

1. Line 66 parenthesis missing at VBN

Parenthesis added.

Reviewer #2

2.1. (38-53) Abstract is written in such a way that it is unclear. At first sight it is not straightforward to grasp which of the findings/methods mentioned already existed (and so serve as a motivation) and which ones refer to the present study. A more consistent use of past continuous and present perfect, as well as, the specification of the actor of some sentences would help clarify this.

2.1. Response: The abstract has been rewritten [32 – 55].

2.2. Introduction could benefit from a final paragraph explaining how the rest of the report is laid out.

2.2. Response: We have added this to the final paragraph [120 - 134].

2.3. (71-72) Comment: This claim seems somewhat stretched. Perhaps rephrase to label it as "one of the most important challenges". Also, the word disbelief seems unsuitable. Perhaps, "difficulty to grasp" or similar would be less of an "accusation".

2.3. Response: Deleted.

2.4. (83-85) Comment: No-one is specifically to blame but current societies have been created based on consumption patterns and culture which results in unsustainable use of resources and generation of environmental impacts. While "no-one" is to blame, we really are all to blame since we all are part of the responsible system. The argument is still valid, however I suggest the authors re-framed along the lines of: this idea (environmental impacts as a consequence of consumption implied by current consumption practices), although common in science, is not easy to make mainstream - therefore perpetuating the effects of its ignorance in individual consumers. See: Reisch, L.A. and Thogersen, J. eds., 2015. Handbook of research on sustainable consumption. Edward Elgar Publishing.

2.4. Response: As per the advice of reviewer 1 (1.1.), the introduction section has now been rewritten and this sentence no longer exists.

2.5. (96-97) It seems to place all of the weight of consumer-cultural change on the consumer. This seems to disregard the fact that the institutional context also informs and affects people's consumption behaviour. While consumer behaviour is important in understanding the factors behind environmental behaviour, consumers alone are unable to change their consumption culture without the right external (institutional) support. Therefore, the abuse of natural resources cannot be attributed solely to the consumer (or their morality). Although a similar argument can still be made, that the degree of engagement in nature-protection is different and this may partly have to do with issues of moral judgements. I suggest that the authors reword it in order to reflect the issue explained above. See: Reisch, L.A. and Thogersen, J. eds., 2015. Handbook of research on sustainable consumption. Edward Elgar Publishing. See: Reisch, L.A. and Thogersen, J. eds., 2015. Handbook of research on sustainable consumption. Edward Elgar Publishing.

2.5. Response: As per the advice of reviewer 1 (1.1.), the introduction section has now been rewritten and this sentence no longer exists.

2.6. (340-341) The choice of participant recruitment platform is justified by claiming that the distribution better reflects the general population. This seems to be assumed, can this be supported with evidence? Also, do the authors mean "the general population of the US" or the global population? This should be coherent with the fact that participants are from the US.

2.6. Response. Explanation added [416 – 422].

2.7. (355 Measures) This section contains some very superficial explanations of some measures for example:

(389-391) "Environmental behaviours": Some examples of the type of items used could be added here to quickly illustrate their nature. Why was this scale chosen for measuring environmental behaviours? This is not trivial and requires some rationale. Another example is

(395-397) "Control Variables": Do the authors mean that gender, age and education are the 3 variables they controlled for? This seems unclear at first glance.

2.7. Response. As per comment 1.10., an explanation for the questionnaire variables has been added [471 – 530].

2.8. The "Results" section looks fine. However, the "Discussion" section seems redundant as it basically just summarises the Results section more comprehensively. Perhaps the "Discussion" section could be streamlined and absorbed into the "Results" section.

2.8. Response: As per comment 1.14, this section has now been placed after the results section [695], and we have added a general discussion section to present the findings in relation to the wider literature [746].

2.9. The "Implications" section is good and, I believe, could just replace the "Discussion" section. However, there is a lack of consideration for limitations of the research. Some examples are: the use of self-reported methods (social desirability bias and potentially hypotheticallity bias), the use of respondents only from the US and how this may affect generalisability of the results and, when measuring the effect of moral philosophies on environmental behaviour, you are effectively measuring an attitude-behaviour relationship which may be subject to similar problems to the TPB that result in the att-bh gap (following the authors' argumentation from lines 320-321).

2.9. Response: Limitations were added [832 – 868].

---

## [Decision Letter · Decision Letter 1]

14 Sep 2020

The effects of idealism and relativism on the moral judgement of social vs. environmental issues, and their relation to self-reported pro-environmental behaviours

PONE-D-20-12480R1

Dear Dr. Zaikauskaite,

We’re pleased to inform you that your manuscript has been judged scientifically suitable for publication and will be formally accepted for publication once it meets all outstanding technical requirements.

Kind regards,

Nikolaos Georgantzis, Dr.

Academic Editor

PLOS ONE

Additional Editor Comments (optional):

Reviewers' comments:

Reviewer's Responses to Questions

**Comments to the Author**

1. If the authors have adequately addressed your comments raised in a previous round of review and you feel that this manuscript is now acceptable for publication, you may indicate that here to bypass the “Comments to the Author” section, enter your conflict of interest statement in the “Confidential to Editor” section, and submit your "Accept" recommendation.

Reviewer #1: All comments have been addressed

Reviewer #2: All comments have been addressed

2. Is the manuscript technically sound, and do the data support the conclusions?

Reviewer #1: Yes

Reviewer #2: Yes

3. Has the statistical analysis been performed appropriately and rigorously? 

Reviewer #1: Yes

Reviewer #2: Yes

4. Have the authors made all data underlying the findings in their manuscript fully available?

Reviewer #1: Yes

Reviewer #2: Yes

5. Is the manuscript presented in an intelligible fashion and written in standard English?

Reviewer #1: Yes

Reviewer #2: Yes

6. Review Comments to the Author

Reviewer #1: The authors have revised the paper significantly. The paper has now improved and I believe could be published.

Reviewer #2: (No Response)

7. PLOS authors have the option to publish the peer review history of their article (what does this mean?). If published, this will include your full peer review and any attached files.

Reviewer #1: No

Reviewer #2: **Yes: **Dimitris Georgantzis Garcia

---

## [Editor Report · Acceptance letter]

15 Oct 2020

PONE-D-20-12480R1 

The effects of idealism and relativism on the moral judgement of social vs. environmental issues, and their relation to self-reported pro-environmental behaviours 

Dear Dr. Zaikauskaite:

I'm pleased to inform you that your manuscript has been deemed suitable for publication in PLOS ONE. Congratulations! Your manuscript is now with our production department. 

Kind regards, 

on behalf of

Prof. Nikolaos Georgantzis 

Academic Editor

PLOS ONE